# 3D Human Pose Estimation from Multiple Dynamic Views via Single-view Pretraining with Procrustes Alignment

## ABSTRACT

3D Human pose estimation from multiple cameras with unknown calibration has received less attention than it should. The few existing data-driven solutions do not fully exploit 3D training data that are available on the market, and typically train from scratch for every novel multi-view scene, which impedes both accuracy and efficiency. We show how to exploit 3D training data to the fullest and associate multiple dynamic views efficiently to achieve high precision on novel scenes using a simple yet effective framework, dubbed *Multiple Dynamic View Pose estimation* (MDVPose). MDVPose utilizes novel scenarios data to finetune a single-view pretrained motion encoder in multi-view setting, aligns arbitrary number of views in a unified coordinate via Procruste alignment, and imposes multi-view consistency. The proposed method achieves 22.1 mm P-MPJPE or 34.2 mm MPJPE on the challenging in-the-wild Ski-Pose PTZ dataset, which outperforms the state-of-the-art method by 24.8% P-MPJPE (-7.3 mm) and 19.0% MPJPE (-8.0 mm). It also outperforms the state-of-the-art methods by a large margin (-18.2mm P-MPJPE and -28.3mm MPJPE) on the EgoBody dataset. In addition, MDVPose achieves robust performance on the Human3.6M datasets featuring multiple static cameras. Code will be released upon acceptance.

## CCS CONCEPTS

• **Computing methodologies** → **Tracking**..

## KEYWORDS

3D human pose estimation, multi-view, dynamic viewpoint

## 1 INTRODUCTION

3D human pose estimation (HPE) [41] is a heated research area in computer vision that estimates the 3D coordinate positions of human body joints, known as 3D pose. It is one of the fundamental techniques used in understanding human behavior analysis and can be applied in many areas such as video surveillance, virtual reality, healthcare, and autonomous driving. Depending on the number of camera views, 3D HPE can be divided into single-view and multi-view 3D HPE.

Multi-view 3D human pose estimation methods [14, 17, 33] typically require precise camera calibration during both training and inference. However, requiring camera calibration have a few issues

*Conference'17, July 2017, Washington, DC, USA*
© 2024 Copyright held by the owner/author(s). Publication rights licensed to ACM.
ACM ISBN 978-x-xxxx-xxxx-x/YY/MM
https://doi.org/10.1145/nnnnnnn.nnnnnnn

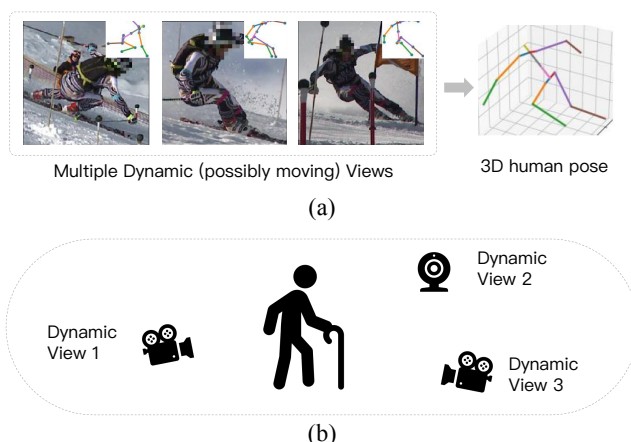

(a)

(b)

**Figure 1: 3D human pose estimation from multiple dynamic (possibly moving) views. (a) Sports capture. (b) Multiple views in a traffic network.**

that limit their application in practice. First, obtaining camera calibration is inconvenient or impractical, especially during inference. Second, due to the high cost of motion capture, the scarcity of 3D training data becomes a bottleneck. Since motion capture data is usually recorded in a constrained space, there is a lack of in-the-wild data [47]. Third, and more importantly, these methods cannot be applied to dynamic (possibly moving) cameras without camera calibration.

Human pose estimation from multiple cameras with unknown calibration has received less attention than it should. For sports capture where close-ups of players are captured in front of moving cameras, camera calibration cannot easily be estimated [37], as shown in Fig. 1(a). Another scenario is in a dynamic camera network for autonomous driving, a pedestrian's pose can be estimated collaboratively using several potentially moving cameras in a traffic Internet of Things (IoT) network, such as vehicle-installed cameras and surveillance cameras (See Fig. 1 (b)).

MetaPose [37] is the first multi-view 3D HPE method that requires no camera parameters at both training and inference time, and thus can be applied to dynamic cameras. MetaPose aggregates pose predictions and uncertainty estimates across multiple views, and outperforms both classical bundle adjustment and weakly-supervised monocular 3D baselines on the Human3.6M dataset, as well as the more challenging in-the-wild Ski-Pose PTZ dataset. *Nevertheless, MetaPose exploits only 2D keypoint data for training and chooses not to make use of 3D joint annotations, whereas we exploit 3D data when available.* FLEX [13] and HMVformer [46] design cross-view deep feature fusion to integrate the relationship between views, to avoid camera calibration. Nonetheless, FLEX [13] and HMVformer [46] train from scratch on every novel scene,



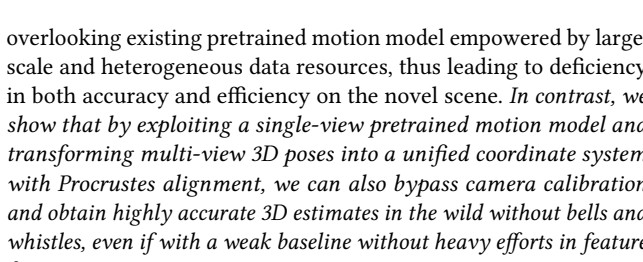

**Figure 2: Errors of 3D human pose estimation from multiple dynamic (possibly moving) views on SkiPose.**

overlooking existing pretrained motion model empowered by large-scale and heterogeneous data resources, thus leading to deficiency in both accuracy and efficiency on the novel scene. *In contrast, we show that by exploiting a single-view pretrained motion model and transforming multi-view 3D poses into a unified coordinate system with Procrustes alignment, we can also bypass camera calibration and obtain highly accurate 3D estimates in the wild without bells and whistles, even if with a weak baseline without heavy efforts in feature fusion.*

We propose the **M**ultiple **D**ynamic **V**iew Pose Estimation (MDV-Pose) framework, to exploit available 3D joint annotations to the fullest and associate different views by human 3D joints themselves. *With a relatively small amount of novel scene training data, MDV-Pose can make highly accurate predictions on novel scenes.* Specifically, 2D keypoint sequences from multiple views are fed into a single-view pretrained motion encoder, which can be adapted to the downstream 3D pose estimation task (essentially 2D-to-3D motion lifting and inpainting). Then, we fine-tune the single-view motion encoder using limited 3D annotations of novel scenarios if they are available, and predict the 3D pose in each view. Subsequently, we associate and procrustes-align all views by the predicted 3D human joints themselves in a unified coordinate, and apply multi-view consistency. By exploiting the powerful pretrained motion encoder combined with transfer learning, and further imposing constraints of multi-view consistency, high accuracy is achieved for novel scenarios such as skiing. Our proposed MDVPose shows a superior performance of 22.1 P-MPJPE or 34.2 MPJPE, which exceeds the state-of-the-art method by 24.8% P-MPJPE (-7.3 mm) and 19.0% MPJPE (-8.0 mm) on the challenging in-the-wild Ski-Pose PTZ dataset [12] (See Fig. 2). It also outperforms the state-of-the-art methods by a large margin (-18.2mm P-MPJPE and -28.3mm MPJPE) on natural scenes in the EgoBody dataset. In addition, MDVPose achieves robust performance on the Human3.6M datasets featuring multiple static cameras even if using a weak baseline without heavy efforts on multi-view feature fusion.

Moreover, MDVPose designs a flexible strategy to train and test on an arbitrary number of dynamic views. In a complicated environment in the wild, the number of views can vary over time. In a skiing motion capture scenario, if the subject skies in long tracks, it is likely that he/she would be out of sight from some

of the camera views. In a traffic IoT network, as the car-mounted cameras move, the pedestrian is probably only available in some of the views. Therefore, we train arbitrary number of dynamic views, enabling the algorithm to adapt to real-world scenarios.

Our contributions are summarized as follows:

- We propose a simple yet effective framework that estimates 3D human pose from multiple dynamic (possibly moving) views which does not require camera calibration, and outperforms state-of-the-art methods by a large margin.
- We utilize a shared pretrained single-view motion encoder to extract multi-view features, followed by exploiting 3D training data of the novel scenario when it is available to fine-tune the novel scene poses. Further, we procrustes-align the multiple dynamic views almost for free in a unified coordinate, and impose multi-view consistency to achieve high accuracy. Experiments on multiple challenging datasets show the effectiveness of our framework in natural scenes.
- We propose a flexible strategy to train and test on arbitrary number of dynamic views, enabling the algorithm to be practical and flexible in real-world scenarios.

## 2 RELATED WORK

**Multi-view 3D HPE.** Supervised multi-view 3D HPE methods [4, 10, 36], including multi-view single person [15] and multi-person [2, 4, 18] methods, can predict highly accurate poses, but typically require precise camera calibration during both training and inference. Most methods [9, 14, 17, 33] obtain the 2D pose by running a CNN over 2D poses given in multiple views. To exploit temporal information, Chen et al. [4] couple cross-view tracking and multi-human 3D pose estimation in a unified framework. TesseTrack [34] handles multi-person 3D body joint reconstruction and association in space and time in a single end-to-end learnable framework. VoxelPose[36] aggregates features of all camera views in the 3D voxel space and directly operates in the 3D space to avoids making incorrect decisions in each camera view. However, supervised methods has a few issues. First, obtaining camera calibration is in convenient or impractical during inference. Second, due to the high cost of motion capture, the scarcity of 3D training data becomes a bottleneck. Third, and more importantly, these methods cannot be applied to dynamic (possibly moving) cameras without camera calibration.

Some works turn to weakly or self-supervised solutions [5, 16, 22, 40]. EpipolarPose uses 2D poses from multi-view images and then utilizes epipolar geometry to obtain a 3D pose and camera geometry, which are subsequently used to train a 3D pose estimator. Iqbal et al. [16] propose a weakly-supervised baseline to predict pixel coordinates of joints and their depth in each view and penalize the discrepancy between rigidly aligned predictions for different views during training. RepNet [39] and Chen et al. [3] use more realistic data to train 2D-to-3D lifting networks. CanonPose [40] presents a self-supervised approach that learns a single image 3D pose estimator from unlabeled multi-view data. These solutions, however, do not allow pose inference from multiple images.

**3D HPE from Multiple Dynamic Views.** Human pose estimation from multiple cameras with unknown calibration has received less attention than it should. MetaPose [37] is the first multi-view

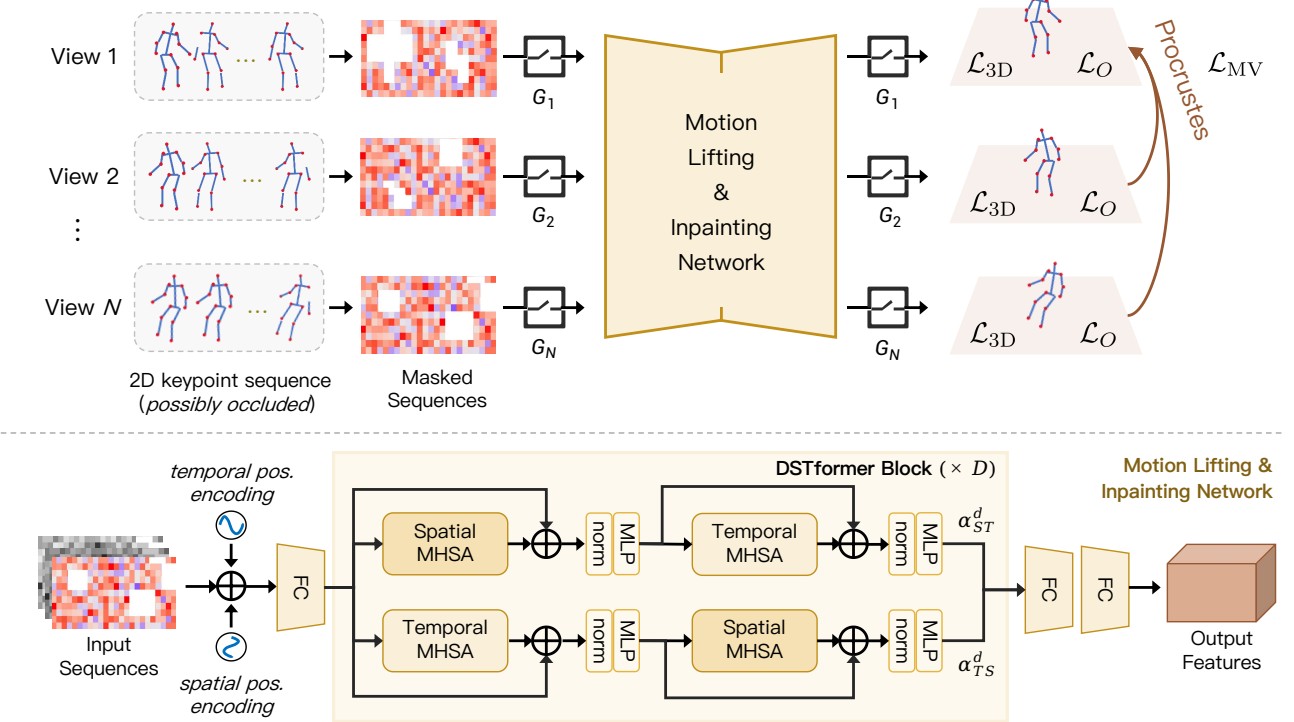

**Figure 3: Overview.** *Top:* **The pose sequence of each view is fed into the motion lifting and inpainting network (Sec. 3.2) though a control gate (Sec. 3.4) and finetuned using 3D data of the novel scene with multi-view consistency (Sec. 3.4, 3.5).** *Bottom:* **The motion lifting and inpainting network we adopt is DSTFormer.**

3D HPE method that requires no camera parameters at both training and inference time, and thus can be applied to dynamic (possibly moving) cameras. MetaPose aggregates pose predictions and uncertainty estimates across multiple views and trains the network without 3D Supervision. The advantage of MetaPose is that it requires only 2D keypoint data for training, which is suitable for cases where 3D data is unavailable. Yet, the downside of MetaPose is it does not utilize the 3D training data when available. FLEX [13] presents an extrinsic parameter-free multi-view model, in the sense that it does not require extrinsic camera parameters. It learns fused deep features through a multi-view fusion layer. HMVformer [46] proposes a hierarchical multi-view fusion transformer framework for 3D HPE, incorporating cross-view feature fusion methods into the spatial and temporal feature extraction process in a coarse-to-fine manner. In contrast, we show that without designs in deep feature fusion, by exploiting a single-view pretrained motion model and transforming multi-view 3D poses into a unified coordinate system with Procrustes alignment which is robust to the relative positions of multiple views, we can obtain highly accurate 3D estimates in the wild.

**3D HPE with Transformer backbones.** 3DHPE with Transfomer backbones, such as PoseFormer [45], MHFormer [24], MixSTE [42], and MotionBERT [48], have gained great success due to its competence in handling sequences. MotionBERT [48] proposes

a pretraining stage in which a motion encoder is trained to recover the underlying 3D motion from noisy partial 2D observations, and implements the motion encoder with a Dual-stream Spatio-temporal Transformer (DSTformer) neural network. Recently, PoseFormerV2 [44] exploits a compact representation of lengthy skeleton sequences in the frequency domain to efficiently scale up the receptive field and boost robustness to noisy 2D joint detection. HSTFormer [32] focuses on capturing multi-level joints' spatial-temporal correlations from local to global gradually for accurate 3D HPE, and [8] focuses on parameter reduction for 3D HPE by leveraging dynamic multi-headed convolutional attention. MotionAGFormer [29] presents an Attention-GCNFormer (AGFormer) block that divides the number of channels by using two parallel Transformer and GCNFormer streams, and reportedly achieves impressive results.

Orthogonal to these efforts in improving the Transformer backbones, we simply take advantage of the Transformer backbone as our motion encoder, and focus on improving the performance and flexibility of 3D HPE from multiple dynamic views. We adopt the DSTFormer proposed by motionBERT [48].

## 3 METHOD

### 3.1 Overview

The overview of the proposed framework is demonstrated in Fig. 3. The pose sequences of multiple views are fed into the shared

motion lifting and inpainting network. We adopt the DSTFormer in motionBERT [48] (Sec. 3.2) as our motion lifting and inpainting network, where inpainting is achieved by randomly masking 2D skeletons during training. Subsequently, we fine-tune the motion lifting and inpainting network using 3D data of the novel scene with multi-view consistency (Sec. 3.3, 3.5). The motion feature and 3D pose output of each view are passed through a control gate (Sec. 3.4) during training to mimic in-the-wild cases where the number of views can vary over time.

## 3.2 DSTformer Backbone for Feature Extraction

Given the input 2D keypoint sequences from $N$ dynamic views $\mathbf{x} \in \mathbb{R}^{N \times T \times J \times C_{in}}$, and 3D joints sequence from multiple dynamic views $\mathbf{X} \in \mathbb{R}^{N \times T \times J \times C_{out}}$, we project the 2D keypoints to a high-dimensional feature $\mathbf{F}^0 \in \mathbb{R}^{N \times 1 \times J \times C_f}$ following motionBERT [48]. $N$ is the number of views, $T$ is the pose sequence length, $J$ is the number of joints in the human skeleton, and $C_{in}$ is the number of input channels. Learnable spatial positional encoding $P_{pos}^T \in \mathbb{R}^{N \times 1 \times J \times C_f}$ and temporal positional encoding $P_{pos}^T \in \mathbb{R}^{N \times T \times 1 \times C_f}$ are then added to the extracted feature. The sequence-to-sequence model DSTformer is then used to calculate $\mathbf{F}^d \in \mathbb{R}^{N \times T \times J \times C_f}$ ($d = 1, ..., D$), where $D$ is the network depth. Same as motionBERT, a linear layer is applied to $\mathbf{F}^D$ to compute the motion feature $\mathbf{E} \in \mathbb{R}^{N \times T \times J \times C_e}$. $C_{in}, C_f, C_e$, and $C_{out}$ are the channel numbers of input, feature, embedding, and output respectively. $C_{out}$ equals 3, i.e. the three dimensions $X, Y, Z$. We let batch size equal the number of viewpoints $N$ during training.

To extract the motion features, the Dual-stream Spatio-temporal Transformer which stacks the spatial and temporal MHSA blocks alternately in two branches [48] and the output features of the two branches are adaptively fused by an attention regressor. The dual-stream-fusion module is repeated for D times:

$$\mathbf{F}^d = \alpha_{ST}^d \circ T_1^d(S_1^d(\mathbf{F}^{d-1})) + \alpha_{TS}^d \circ T_2^d(S_2^d(\mathbf{F}^{d-1})), \quad (1)$$

where $d \in 1, ..., D$ is the network depth and $\mathbf{F}^d$ denotes the feature embedding at depth $d$. $S$ and $T$ are spatial and temporal blocks. Adaptive fusion weights $\alpha_{ST}^d, \alpha_{TS}^d \in \mathbb{R}^{D \times N \times T \times J}$ are given by a shallow layer, following the same architecture as in motionBERT [48].

The DSTFormer architecture is illustrated in Fig. 3. The building blocks of DSTFormer consists of spatial and temporal *Multi-Head Self Attention* (MHSA) blocks stacked in different orders, forming two parallel computation streams which are expected to specialize in different spatial-temporal aspects. DSTFormer takes multi-view 2D keypoint sequences $\mathbf{x}$ as input and passes it through a fully connected (FC) later before feeding it into the DSTFomer.

**Spatial MHSA**. The *Spatial MHSA* (S-MHSA) models the relationship among the joints from a view within the same time step. Note that each view is computed separately. It is defined as:

$$\text{S} - \text{MHSA}(\mathbf{Q}_S, \mathbf{K}_S, \mathbf{V}_S) = [h_1, ...h_H] \mathbf{W}_S^P,$$
$$h_h = \text{Softmax}\left(\frac{Q_S^h (K_S^h)^\top}{\sqrt{d_K}}\right) \mathbf{V}_S^h, \quad (2)$$

where $\mathbf{W}_S^P$ is a projection parameter matrix. Query $Q_S^h$, key $K_S^h$, and value $V_S^h$ are per frame spatial feature from input $F_S$ from

each attention head $h$. $d_K$ is the feature dimension of $KS$. Residual connection and layer normalization (LayerNorm) are used to the S-MHSA block following classical Transformers [38].

**Temporal MHSA**. The *Temporal MHSA* (T-MHSA), on the other hand, models the relationship across the time steps for a body joint and applies to a feature $F_T \in \mathbb{R}^{T \times C_e}$. The notation is similar to the spatial MHSA.

$$\text{T} - \text{MHSA}(\mathbf{Q}_T, \mathbf{K}_T, \mathbf{V}_T) = [h_1, ...h_H] \mathbf{W}_T^P$$
$$h_h = \text{Softmax}\left(\frac{\mathbf{Q}_T^h (K_T^h)^\top}{\sqrt{d_K}}\right) \mathbf{V}_T^h. \quad (3)$$

## 3.3 Multi-view 3D HPE Finetuning

We use the pretrained motion encoder given by motionBERT [48], where the learned feature embedding $\mathbf{E}$ serves as a 3D-aware and temporal-aware human motion representation. Since the motion encoder adopts masking strategy during pretraining, it can realize pose track inpainting as well. Our work focuses on the 3D human pose estimation task; therefore, we fine-tune the network to achieve 2D-to-3D lifting and inpainting, i.e. 3D human pose estimation.

The loss for multi-view 3D HPE finetuning is computed as:

$$\mathcal{L} = \sum_{n=1}^{N} \lambda_{3D} \mathcal{L}_{3D} + \lambda_O \mathcal{L}_O \quad (4)$$

where $N$ is the number of viewpoints. $\mathcal{L}_{3D}$ is the pose regression loss, $\mathcal{L}_O$ is the velocity loss. Predicted velocity is calculated by $\mathbf{O}^t = \mathbf{X}^t - \mathbf{X}^{t-1}$, and likewise for ground truth velocity $\hat{\mathbf{O}}^t$.

$$\mathcal{L}_{3D} = \sum_{t=1}^{T} \sum_{j=1}^{J} \left\| \mathbf{X}_j^t - \hat{\mathbf{X}}_j^t \right\|,$$
$$\mathcal{L}_O = \sum_{t=1}^{T} \sum_{j=1}^{J} \left\| \mathbf{O}_j^t - \hat{\mathbf{O}}_j^t \right\|, \quad (5)$$

where $J$ is the number of joints, and $T$ the sequence length.

## 3.4 Arbitrary Number of Viewpoint

We design the framework such that the model can take a arbitrary number of viewpoints as input. This is done by gating the data of any viewpoint with a certain probability during training, only keeping some of the viewpoints. Let $\mathbf{E}_i \in \mathbb{R}^{T \times J \times C_{out}}$ be the motion feature and $\mathbf{X}_i \in \mathbb{R}^{T \times J \times C_{out}}$ be the output from View $i$, we add a control gate $G_i \in 0, 1$ at each view to mask the feature and 3D pose output sequence.

$$\tilde{\mathbf{E}}_i = G_i * \mathbf{E}_i,$$
$$\tilde{\mathbf{X}}_i = G_i * \mathbf{X}_i, \quad (6)$$

When $G_i = 1$, View $i$ is valid, otherwise it is invalid. $P_i$ is the probability of having $i$ valid views.

$$P_i = \begin{cases} P_1, & i = 1 \\ \frac{1-P_1}{N-1} & i = 2, ..., N \end{cases} \quad (7)$$

## 3.5 Multi-view Consistency

When there are multiple viewpoints in the input, we associate multiple viewpoints by 3D human joints themselves and apply

multi-view consistency constraint. This process can be done almost for free. The multi-view consistency loss. $\mathcal{L}_{MV}$ is computed as follows.

(1) We predict the 3D human pose of each viewpoint $i$ at time $t$ as $\mathbf{X}_i^t \in \mathbb{R}^{J \times C_{out}}$. $C_{out} = 3$.

(2) Using the first viewpoint as the reference view, we procrustes-align the 3D pose estimated from the other viewpoints to the first viewpoint if there should be other viewpoints, i.e. $N \geq 2$. PROC$(\mathbf{x}, \mathbf{y})$ denotes the procrustes alignment of 3D joints.

$$\text{PROC}(\mathbf{x}, \mathbf{y}) = s(\mathbf{R}\mathbf{x} + \mathbf{t})$$

$$s.t. \min_{s, \mathbf{R}, \mathbf{t}} \frac{1}{J} \sum_{j=1}^{J} \left\| \mathbf{y}_j - s(\mathbf{R}\mathbf{x}_j + \mathbf{t}) \right\| \quad (8)$$

The procrustes alignment calculates the scale $s$, rotation $\mathbf{R}$, and translation $\mathbf{t}$ between two sets of 3D point correspondences, which is the same as calculating P-MPJPE from MPJPE [41].

$$\tilde{\mathbf{X}}_i^t = \text{PROC}(\mathbf{X}_i^t, \mathbf{X}_1^t), i = 2, ..., N \quad (9)$$

(3) Using the procruste-aligned $\tilde{\mathbf{X}}_i^t$, compute the multi-view consistency loss:

$$\mathcal{L}_{MV} = \sum_{i=2}^{N} \left\| \tilde{\mathbf{X}}_i^t - \mathbf{X}_1^t \right\| \quad (10)$$

Thus, the overall loss of our framework becomes:

$$\mathcal{L} = \mathbb{I}_{(G_i=0)(i=2,...,N)} \lambda_{MV} \mathcal{L}_{MV} + \sum_{i=1}^{N} (\lambda_{3D} \mathcal{L}_{3D} + \lambda_O \mathcal{L}_O), \quad (11)$$

where $N$ is the number of viewpoints. $\mathbb{I}$ is a binary indicator function to decide whether to include the multi-view consistency loss.

## 4 EXPERIMENTS

### 4.1 Experimental Settings

**Data.** We evaluate the performance of our proposed method on three 3D human pose estimation benchmarks, i.e., one challenging in-the-wild dataset SkiPose [12], one recent dataset EgoBody [43], and one well-established dataset Human3.6M [15]. **Ski-Pose PTZ (SkiPose)** is a challenging in-the-wild dataset with six *moving* pan-tilt-zoom cameras that is perfect for evaluating our proposed method. This multi-view pant-tilt-zoom-camera (PTZ) dataset features competitive alpine skiers performing giant slalom runs. It provides labels for the skiers' 3D poses in each frame, their projected 2D pose in all 20k images, and accurate per-frame calibration of the PTZ cameras.The model is trained and tested using an official training and testing set. **EgoBody** is another large-scale dataset for 3D human motions during social interactions in complex 3D scenes. For each sequence, multiple Azure Kinects capture the two-subject interactions from different views with RGBD streams, and a synchronized HoloLens2 worn by one subject captures the first-person view image. We only use the RGB data from Kinect cameras for training in our experiments. **Human3.6M** is a large-scale 3D dataset of 3.6 Million accurate 3D Human poses, acquired by recording the performance of 5 female and 6 male subjects, under 4 different viewpoints, providing a diversity of human activities.We keep the same training and test split as in [28, 48].

**Metrics.** We report (1) Mean Per Joint Position Error (MPJPE), and (2) Procrustes aligned Mean Per Joint Position Error (P-MPJPE) that measure the L2-error of 3D joint estimates after applying the optimal rigid alignment (including scale) [7, 41]. Some other methods report (3) normal mean per joint position error (NMPJPE) that normalizes MPJPE to exclude the effect of scale.

**Implementation details.** We use 2 RTX3090 to train our model. DSTformer is trained with depth $D = 5$, number of heads $h = 8$, feature size $C_f = 512$, embedding size $C_e = 512$, following motionBERT. The probability of having 1 valid viewpoint during training is set to $P_1 = 0.5$. $\lambda_{MV} = 1, \lambda_{3D} = 1, \lambda_O = 1$. We set batch size to be the number of views $N$, which is 6 for SkiPose and 4 for Human3.6M. For Human3.6M, the 2D skeletons are provided by 2D pose estimator trained on MPII [1] and Human3.6M following the common practice of [48].

**Table 1: Comparisons with SOTA methods on SkiPose. The results of AniPose, Rhodin et al. [35], CannonPose[40], MetaPose[37] were provided by [37]. IR means iterative refinement, the details of which can be found in motionBERT. ↓ means the lower the better. Best in bold.**

|  |  | MPJPE ↓ | PMPJPE ↓ | NMPJPE↓ |
|---|---|---|---|---|
| AniPose w/ GT[20] | Cell Rep.'21 | - | 50 | 221 |
| CanonPose [40] | CVPR'21 | 128 | 90 | - |
| Chen [6] | AAAI'21 | 99.4 | 74.7 | - |
| MetaPose [37] | CVPR'22 | - | 42 | 53 |
| MetaPose (IR) [37] | CVPR'22 | - | 30 | 53 |
| FLEX [13] | ECCV'22 | 65.5 | - | - |
| HMVformer [46] | MM'23 | 42.2 | 29.4 | - |
| MDVPose (Ours) |  | **34.2** | **22.1** | - |

**Table 2: Comparisons with SOTA methods on EgoBody. The results of SPIN [23], METRO [25], PARE [21] and EFT [19] were provided by the official repository of EgoBody [43]. ft denotes results of fine-tuning SPIN, METRO and EFT on the EgoBody training set. ↓ means the lower the better. gt indicates ground truth 2D keypoints were fed into the network. Best in bold.**

|  |  | MPJPE ↓ | PMPJPE ↓ |
|---|---|---|---|
| PARE [21] | ICCV'21 | 123.0 | 83.8 |
| SPIN (ft) [23] | ICCV'19 | 106.5 | 67.1 |
| METRO (ft) [25] | CVPR'21 | 98.5 | 66.9 |
| EFT (ft) [19] | 3DV'20 | 102.1 | 64.8 |
| MEEV [30] | ECCV'22 | 82.3 | 55.1 |
| MDVPose (Ours) |  | **54.0** | **36.9** |
| MDVPose (gt) (Ours) |  | **36.6** | **24.2** |

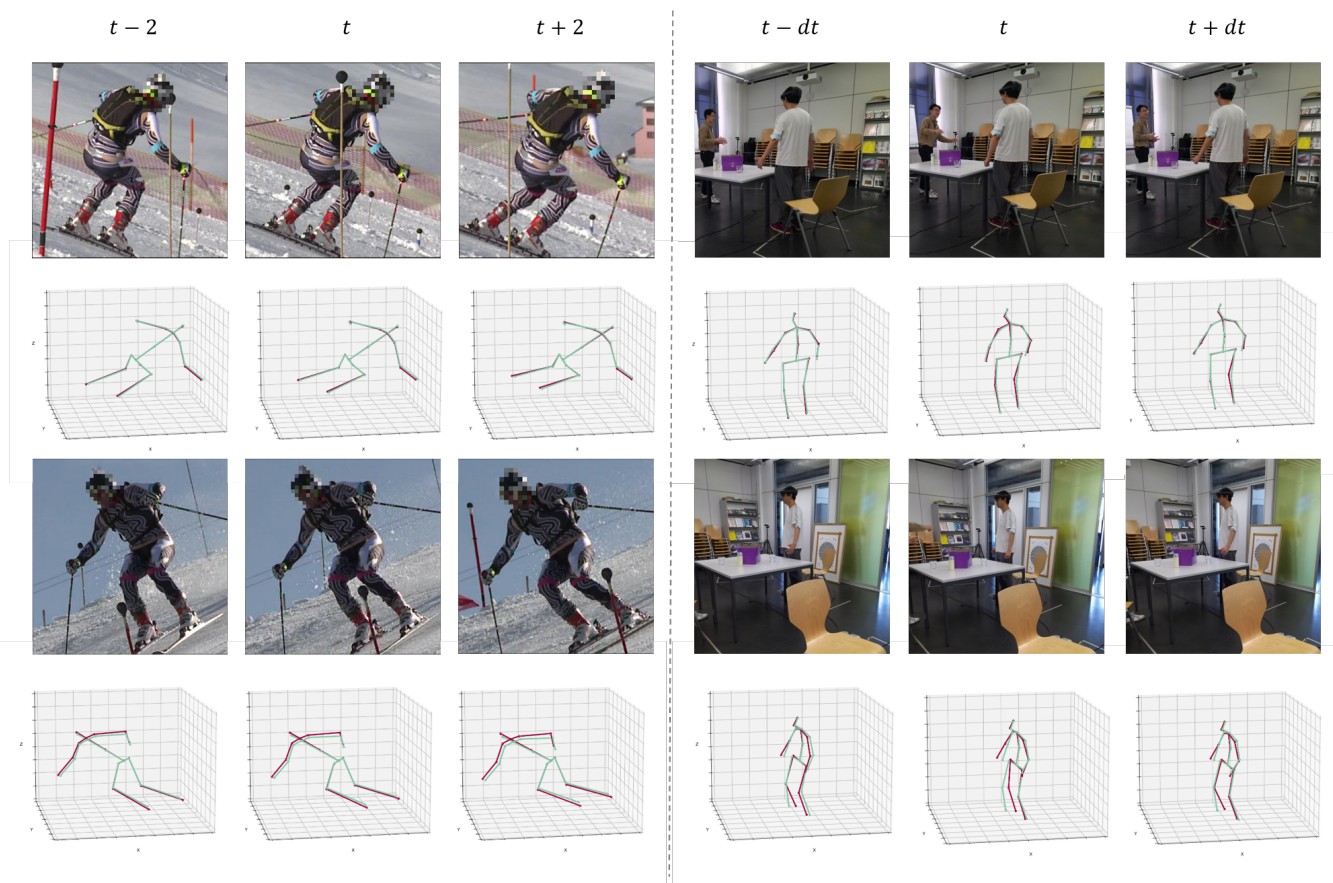

**Figure 4: Qualitative Evaluation on the SkiPose (left) and EgoBody (right) dataset. Output 3D pose sequences of synchronized dynamic views. The red skeletons are the ground truth (GT) and the green skeletons are the predicted poses. The predicted poses align well with the ground truth.**

## 4.2 Quantitative Evaluation

**Quantitative Evaluation on SkiPose.** We compare our method with the state-of-the-art methods for 3D HPE using multiple dynamic views on the SkiPose dataset, and the results are listed in Table 1. Our method achieves an unprecedented 22mm P-MPJPE (after procrustes alignment) as well as 34mm MPJPE. The P-MPJPE(↓) is 7.4mm (25.2%) lower than the second-best model HMVPose [46] (22mm vs. 29.4mm). Note that the MetaPose performance was achieved with a series of iterative refinement steps, which we do not perform. Even if we use a weak baseline without heavy efforts in feature fusion, we were able to easily achieve highly accurate pose predictions, showing the efficacy of the proposed method on challenging outdoor data

**Quantitative Evaluation on EgoBody.** Table 2 presents the quantitative evaluations on the EgoBody dataset, where two subjects interact in natural scenes. We first evaluate the results when an off-the-shelf 2D keypoint detector [11] is used during reference. The proposed method significantly outperforms the state-of-the-art

methods on EgoBody, surpassing the EgoBody Challenge champion MEEV [30] by 18.2mm P-MPJPE and 28.3mm MPJPE. Subsequently, we show the upper bound of our method when ground truth 2D keypoints are fed into the network during inference, listed as "MDV-Pose(gt)". The experimental results show that our method produces highly accurate 3D pose estimates for novel natural videos without bells and whistles.

**Quantitative Evaluation on Human3.6M.** Table 3 presents the quantitative results on Human3.6M. The proposed MDVPose achieves state-of-the-art accuracy on the competitive benchmark, proving that the MDVPose is robust and generalizes well. Note that our method cannot be compared with HMVPose or FLEX because of two reasons. First, HMVPose and FLEX are not intrinsic-parameter free on Human3.6M, whereas our MDVPose uses neither intrinsic parameters nor extrinsic parameters. Second, the input 2D keypoints of HMVPose were given by CPN while ours were inherited from motionBERT [48]. It is also worth mentioning that 3D HPE from static cameras are not the focus of this paper; yet, MDVPose

**Table 3: Quantitative comparison of 3D human pose estimation on Human3.6M. Numbers are MPJPE ↓ (mm) when using detected 2D keypoint sequences as inputs, which are provided by 2D pose estimator trained on MPII [1] and Human3.6M following [48].**

| Protocol 1 (MPJPE ↓) | | Dire. | Disc. | Eat | Greet | Phone | Photo | Pose | Purch. | Sit | SitD | Smoke | Wait | WalkD | Walk | WalkT | Avg. |
|---|---|---|---|---|---|---|---|---|---|---|---|---|---|---|---|---|---|
| *Single-view methods* | | | | | | | | | | | | | | | | | |
| VideoPose3D [31] | CVPR'19 | 45.2 | 46.7 | 43.3 | 45.6 | 48.1 | 55.1 | 44.6 | 44.3 | 57.3 | 65.8 | 47.1 | 44.0 | 49.0 | 32.8 | 33.9 | 46.8 |
| Liu et al. [26] | CVPR'20 | 41.8 | 44.8 | 41.1 | 44.9 | 47.4 | 54.1 | 43.4 | 42.2 | 56.2 | 63.6 | 45.3 | 43.5 | 45.3 | 31.3 | 32.2 | 45.1 |
| PoseFormer [45] | ICCV'21 | 41.5 | 44.8 | 39.8 | 42.5 | 46.5 | 51.6 | 42.1 | 42.0 | 53.3 | 60.7 | 45.5 | 43.3 | 46.1 | 31.8 | 32.2 | 44.3 |
| CanonPose [40] | CVPR'21 | - | - | - | - | - | - | - | - | - | - | - | - | - | - | - | 74.3 |
| MHFormer [24] | CVPR'22 | 39.2 | 43.1 | 40.1 | 40.9 | 44.9 | 51.2 | 40.6 | 41.3 | 53.5 | 60.3 | 43.7 | 41.1 | 43.8 | 29.8 | 30.6 | 43.0 |
| MixSTE [42] | CVPR'22 | 36.7 | 39.0 | 36.5 | 39.4 | 40.2 | 44.9 | 39.8 | 36.9 | 47.9 | 54.8 | 39.6 | 37.8 | 39.3 | 29.7 | 30.6 | 39.8 |
| MotionBERT [48] | CVPR'22 | 36.1 | 37.5 | 35.8 | 32.1 | 40.3 | 46.3 | 36.1 | 35.3 | 46.9 | 53.9 | 39.5 | 36.3 | 35.8 | 25.1 | 25.3 | 37.5 |
| PoseFormerV2 [44] | CVPR'23 | - | - | - | - | - | - | - | - | - | - | - | - | - | - | - | 45.2 |
| *Multi-view methods, camera parameters are **given*** | | | | | | | | | | | | | | | | | |
| He [14] | CVPR'20 | 25.7 | 27.7 | 23.7 | 24.8 | 26.9 | 31.4 | 24.9 | 26.5 | 28.8 | 31.7 | 28.2 | 26.4 | 23.6 | 28.3 | 23.5 | 26.9 |
| Qiu [33] | ICCV'19 | 24.0 | 26.7 | 23.2 | 24.3 | 24.8 | 22.8 | 24.1 | 28.6 | 32.1 | 26.9 | 31.0 | 25.6 | 25.0 | 28.1 | 24.4 | 26.2 |
| Iskakov [17] | ECCV'18 | 19.9 | 20.0 | 18.9 | 18.5 | 20.5 | 19.4 | 18.4 | 22.1 | 22.5 | 28.7 | 21.2 | 20.8 | 19.7 | 22.1 | 20.2 | 20.8 |
| *Multi-view methods, uncalibrated cameras, intrinsic and extrinsic camera parameters are **not given*** | | | | | | | | | | | | | | | | | |
| Luvizon [27] | IJCV'22 | 40.0 | 36.0 | 44.0 | 39.0 | 44.0 | **42.0** | 41.0 | 66.0 | 70.0 | 46.0 | 49.0 | 43.0 | **34.0** | 46.0 | 34.0 | 45.0 |
| MDVPose (Ours) | | **35.3** | 37.0 | 37.5 | 31.9 | 39.2 | 45.7 | 36.5 | 33.5 | 48.7 | 54.0 | 39.5 | 36.8 | 34.9 | 24.4 | 25.1 | 37.3 |
| **Protocol 2 (P-MPJPE↓)** | | Dire. | Disc. | Eat | Greet | Phone | Photo | Pose | Purch. | Sit | SitD | Smoke | Wait | WalkD | Walk | WalkT | Avg. |
| *Single-view methods* | | | | | | | | | | | | | | | | | |
| MixSTE [42] | CVPR'22 | 30.8 | 33.1 | 30.3 | 31.8 | 33.1 | 39.1 | 31.1 | 30.5 | 42.5 | 44.5 | 34.0 | 30.8 | 32.7 | 22.1 | 22.9 | 32.6 |
| PoseFormerV2 [44] | CVPR'23 | - | - | - | - | - | - | - | - | - | - | - | - | - | - | - | 35.6 |
| *Multi-view methods, uncalibrated cameras, intrinsic and extrinsic camera parameters are **not given*** | | | | | | | | | | | | | | | | | |
| MetaPose [37] | CVPR'22 | - | - | - | - | - | - | - | - | - | - | - | - | - | - | - | 32.0 |
| MDVPose (Ours) | | 29.7 | 31.4 | 32.2 | 27.1 | 32.9 | 37.3 | 30.0 | 29.4 | 40.9 | 48.3 | 34.3 | 30.3 | 30.1 | 21.2 | 21.8 | **31.8** |

demonstrated robust performance on 3D HPE from multiple views in general.

## 4.3 Qualitative Evaluation

Fig. 4 illustrates several examples from the SkiPose (left) and the EgoBody (right) dataset. Result 3D pose sequences of synchronized views at different time steps are plotted. For SkiPose, the 2D keypoints are provided by the dataset, following previous works FLEX [13] and HMVFormer [46]. For EgoBody, the input 2D keypoints are extracted by an off-the-shelf 2D keypoint detector [11]. As shown in Fig. 4, the predicted 3D poses (green skeletons) all align well with the ground truth (red skeletons).

Fig. 5 compares MDVPose output with the original motionBERT output. Both using the DSTFormer as backbone, our MDVFormer performs significantly better than motionBERT.

## 4.4 Ablations

**Effect of multi-view consistency loss.** We conduct ablation studies on the SkiPose dataset with multiple dynamic views. The Performance with the multi-view consistency loss $\mathcal{L}_{MV}$ versus without $\mathcal{L}_{MV}$ are listed in Table 1. The model with multi-view consistency loss outputs -3 mm lower P-MPJPE, which is more important in analyzing the behavior of the individual human. This is reasonable because the multi-view consistency loss (Eq. 10) basically penalizes P-MPJPE.

**Table 4: Ablation study for multi-view consistency loss on SKiPose. Notation consistent with Table 1.**

| Method | Setting | MPJPE ↓ | PMPJPE ↓ |
|---|---|---|---|
| Ours w/ $\mathcal{L}_{MV}$ | scratch | 39.6 | 31.6 |
| Ours w/o $\mathcal{L}_{MV}$ | fine-tune | **33.7** | 25.0 |
| Ours w/ $\mathcal{L}_{MV}$ | fine-tune | 34.2 | **22.1** |

**Training from scratch versus fine-tuning.** The results trained from scratch using SkiPose training data is reported in Row 1, Table 1. As shown, the fine-tuning strategy outperforms the train-from-scratch strategy by a large margin, proving that mining 3D training data to the fullest can yield the best results. In addition, we plot the error curves of MPJPE and P-MPJPE in Fig. 6. The fine-tuned model converges significantly faster than that of the model trained from scratch, making it more efficient and practical for real-world applications.

**Multi-view versus single view.** We compare the qualitative results using the original motionBERT model and our MDVPose model in Fig. ??. Using the same backbone, MDVPose shows superior performance than motionBERT on the SKiPose dataset.

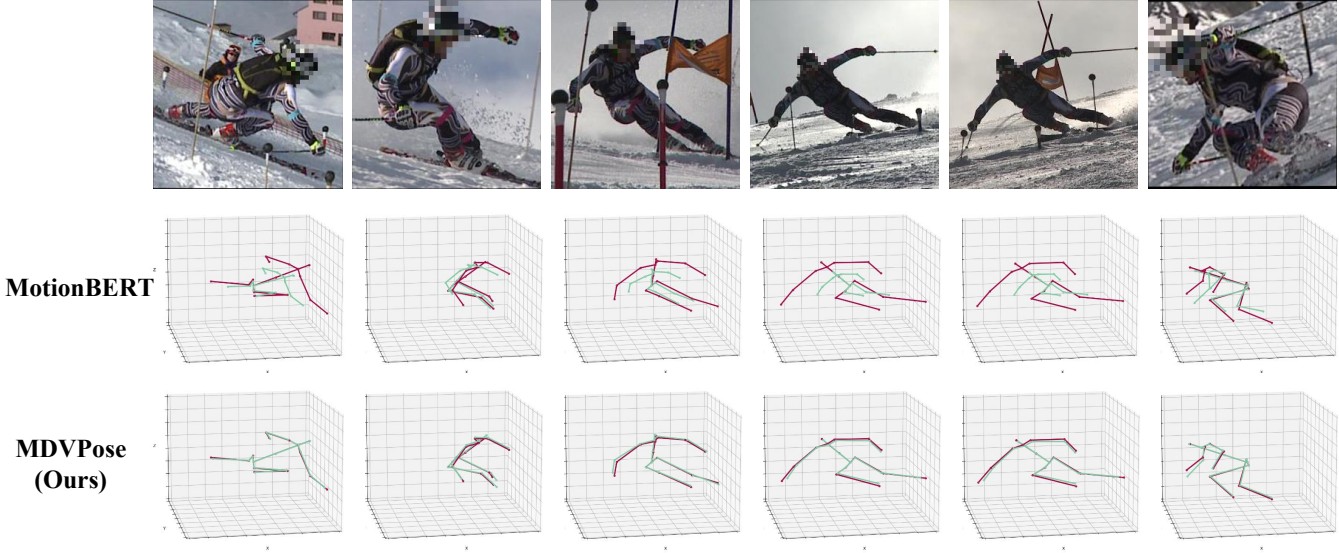

MotionBERT

MDVPose
(Ours)

**Figure 5: Qualitative comparisons on the Ski-Pose PTZ-Camera dataset. Qualitative comparison with state-of-the-art method motionBERT. The red skeletons are the ground truth (GT) and the green skeletons are the predicted poses.**

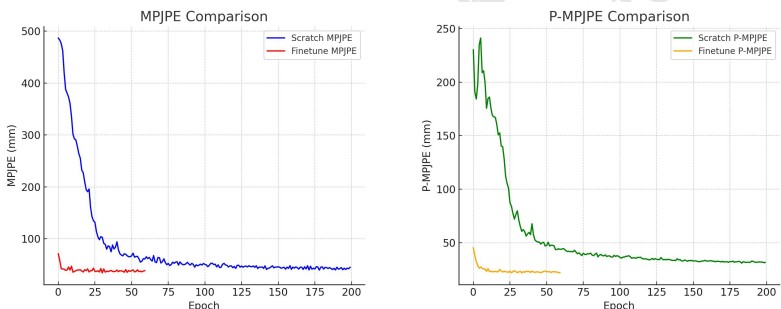

**Figure 6: Fine-tuning vs. training from scratch. Learning curves of fine-tuning versus training from scratch, in terms of MPJPE (left) and P-MPJPE (right).**

## 5 CONCLUSIONS

This paper presents a simple yet effective and flexible framework, MDVPOse for 3D human pose estimation in the wild. MDVPose tackles the problem of estimating 3D coordinates of human joints from RGB images captured using an arbitrary number of dynamic (potentially moving) cameras with unknown positions, orientations, and intrinsic parameters. It utilizes a single-view pretrained Transformer-based motion encoder trained with massive data, and exploits novel scenarios data to finetune the pretrained model before aligning multiple views in a unified coordinate and imposing multi-view consistency. A strategy is designed to train using a arbitrary number of dynamic views, allowing the algorithm to flexibly adapt to real-world scenarios. The proposed framework leads to highly accurate 3D human pose estimates on the challenging Ski-Pose PTZ dataset, outperforming the second-best model by 24.8% in terms of P-MPJPE (-7.3 mm) and 19.0% MPJPE (-8.2 mm). It also outperforms the state-of-the-art methods by a large margin

(-18.2mm P-MPJPE and -28.3mm MPJPE) on the EgoBody dataset. In summary, the benefit of the proposed method are three-folds: first, With a relatively small amount of novel scene training data, MDVPose can make highly accurate predictions on the novel scenes with multiple dynamic views. Second, the proposed simple yet effective framework does not demand camera calibration, making it extremely practical for scenarios where camera calibration is hard, such as skiing. Third, MDVPose adapts to an arbitrary number of dynamic views to be practical and flexible in complicated real-world scenarios.

**Limitations and future work.** MDVPose only uses a weak baseline without heavy efforts on multi-view feature fusion. We believe that with stronger backbones and more designs in cross-view relations the accuracy could be further improved. Currently, MDVPose can only handle single person 3D HPE without considering multi-person tracking. The framework can be extended to multi-person 3D HPE using multiple dynamic views in future, which is a further step towards analyzing human behaviors in the wild.

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

Received 8 April 2024

