# OpenReview forum: "3D Human Pose Estimation from Multiple Dynamic Views via Single-view Pretraining with Procrustes Alignment"
_acmmm.org/ACMMM/2024/Conference — MM2024 Poster_

### Official Review · Reviewer_sYXa · 2024-05-24

**Rating:** 4
**Confidence:** 3

**Summary:**

The paper proposes a simple yet effective framework that estimates 3D human pose from multiple dynamic (possibly moving) views which does not require camera calibration, and outperforms state-of-the-art methods by a large margin. A shared pretrained single-view motion encoder is designed to extract multi-view features, followed by exploiting 3D training data of the novel scenario when it is available to fine-tune the novel scene poses. Further, the paper procrustes-aligns the multiple dynamic views almost for free in a unified coordinate, and imposes multi-view consistency to achieve high accuracy. Experiments on multiple challenging datasets show the effectiveness of our framework in natural scenes.

**Strengths:**

1. This paper is novelty, and the DSTformer effectively enhance the model's performance by spatial-temporal details.
2. Comprehensive comparisons to competing methods.
3. Although uncalibrated cameras, intrinsic and extrinsic camera parameters are not given, the performance is still excellent. This demonstrates the effectiveness and applicability of the method.

**Limitations:**

1. The visualization experiments are insufficient, and the existing multi-view methods are lacking in the comparative analysis.
2. There are errors in the figure citations in this paper; please check carefully.
3. The design advantages of the dual-branch network lack visual experimental validation.
4. There is a lack of analysis regarding the computational complexity of the dual-branch network.

**Suitability:**

2

---

### Official Review · Reviewer_KTZQ · 2024-05-25

**Rating:** 4
**Confidence:** 1

**Summary:**

The paper proposes MDVPose, which is a framework for 3D human pose estimation from multiple dynamic cameras with unknown calibration. While the baseline models either exploits only 2D keypoint data or does not utilize pre-trained motion models, MDVPose successfully avoid these. It leverages Procrustes alignment to transform multi-view 3D poses into a unified coordinate system and exploits single-view pre-trained motion model. By doing so, MDVPose not only obtains highly accurate 3D estimation in the wild but also bypasses the camera calibration which is impractical. It outperforms all stated baseline models both qualitatively and quantitatively in 3 different datasets.

**Strengths:**

- Capable of leveraging existing 3D annotations and pre-trained motion model encoder of motionBERT.
- Achieves highly accurate predictions on novel scenes even with relatively smaller novel scene data.
- Capable of handling arbitrary number of viewpoints
- Extensive experimental results with SkiPose, EgoBody, and Human3.6M, comprehensively show MDVPose exceeds state-of-the-art baselines in both in-the-wild and natural settings.
- Supplementary videos present temporally consistent prediction quality.

**Limitations:**

- What is the reason for adding both spatial positional encoding and temporal positional encoding? Does it perform the best compared to other operations like concatenation?
- Since I am not an expert on this task, I will refer to other reviewers' comments for my final decision.

Minor comment:
- L810: Incorrect reference to a Figure.

**Suitability:**

3

---

### Official Review · Reviewer_6SM3 · 2024-05-27

**Rating:** 3
**Confidence:** 3

**Summary:**

This paper proposes a new muti-view 3d pose estimation method that utilizes a pre-trained backbone with a new control gate design for extending to arbitrary views without camera calibrations.

**Strengths:**

1.	The proposed method got SOTA performance on three benchmark datasets.
2.	The proposed method can be applied to arbitrary views.

**Limitations:**

1. What's the relation between Pi and Gi according to Eq. 7.
2. According to Eq. 7, as N increases, Pi gradually decreases. What is the logic of this design? Later views have less influence?
3. In L752: Fig.?? L854 Capital O should be o.
4. In Table 4, when multi-view consistency loss is added, MPJPE increases in fine-tuning setting. Does this happen on other datasets?
5. Any visualization analysis on views that not be selected? Why are they dropped by the network?

**Suitability:**

2

---

### Meta-Review · Area_Chair_VEqY · 2024-07-05

**Recommendation:** Accept (Poster)
**Confidence:** 4

**Metareview:**

This paper addresses the problem of 3D Human pose estimation from multiple cameras with unknown calibration.
The proposed approach is novel and technically sound, it achieves competitive results, it generalizes to arbitrary views, it is well presented.
The only shortcome is the computationally complexity that is much higher with respect to competitive approaches. However, the computational time is reasonable (also considering the limited computational resources used by the authors) and wouldn't hampers its use real applications, where it is very common to have uncalibrated cameras.
I recommend acceptance.